# Cross-Resistance Pattern and Genetic Studies in Spirotetramat-Resistant Citrus Red Mite, *Panonychus citri* (Acari: Tetranychidae)

Jinfeng Hu [1], Jun Wang [1], Yun Yu [2], Wenhua Rao [1], Feng Chen [1], Changfang Wang [1] and Guocheng Fan [1,*]

1 Fujian Engineering Research Center for Green Pest Management, Key Laboratory for Monitoring and Integrated Management of Crop Pests, Institute of Plant Protection, Fujian Academy of Agricultural Sciences, Fuzhou 350013, China; hujinfeng007@sina.com (J.H.); archy001@126.com (J.W.); raowenhuafj@126.com (W.R.); fchenfj@126.com (F.C.); wangchangfang@faas.cn (C.W.)
2 Technology Center of Fuzhou Customs District, Fuzhou 350001, China; yuyunfzhg@126.com
* Correspondence: guochengfan@126.com

**Abstract:** In the laboratory, an acaricide-susceptible strain of the citrus red mite, *Panonychus citri* (McGregor) (LS-FJ), was used to screen for resistance to spirotetramat. A spirotetramat-resistant strain (ST-NK) obtained after continuous selections through 15 selection cycles (45 generations) exhibited 1668.4-fold greater resistance when compared to the parent generation. Instability of the spirotetramat resistance in the mites was observed during 11 months under spirotetramat-free laboratory conditions. Cross-resistance to spirodiclofen and spiromesifen was detected both in eggs and larvae, but not to five other tested acaricides. Probit lines for F1 heterozygous progeny indicated that the resistance to spirotetramat in the mites was autosomal with neither sex linkage nor maternal effects. The degrees of dominance were 0.15 and 0.23 for the diploid F1 of LS-FJ♀× ST-NK♂and ST-NK♀× LS-FJ♂, and 0.07 and 0.13 for haploid F2 of LS-FJ♀× ST-NK♂and ST-NK♀× LS-FJ♂, respectively, which indicated that the resistance was incompletely dominant. The χ2 analyses from the response of a backcross of crossed F1 progeny and ST-NK and F2 progeny showed that multiple genes are responsible for resistance to spirotetramat.

**Keywords:** *Panonychus citri*; spirotetramat; resistance selection; cross-resistance; genetic analysis

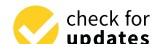



## 1. Introduction

China is the world's largest citrus producer, with 2.8 million hectares of planted area and 41.4 million tons of production [1]. The citrus red mite, *Panonychus citri* (McGregor) (Acari: Tetranychidae), is a key citrus pest that is mainly active in spring and autumn. Citrus red mites' heavy feeding on leaves causes defoliation, twig dieback, fruit drop, and even diminished health and production. Citrus growers in the country mainly spray acaricides to kill the pests [2]. Approximately 1319 acaricide products have been registered for use in pest management in China, of which 53 products contained spirotetramat [3].

Spirotetramat (Movento®), a member of the ketoenol chemical class [4], is a fully systemic pesticide discovered by Bayer CropScience. As a lipid biosynthesis inhibitor, the compound targets a wide range of sap-sucking pests, such as aphis, mealybugs, scales, and psyllids. It is applied on citruses, pome fruit, grapes, mangoes, and cotton plants, among many others [5]. Furthermore, similar to spirodiclofen (Envidor®) and spiromesifen (Oberon®), the compound also shows excellent activity against tetranychid mites [6]. Movento® (spirotetramat) was first registered in 2010 for scale insect management, but the citrus red mite was added to the label in 2016. Movento® has shown an outstanding performance against citrus red mites and has been widely applied by Chinese citrus farmers because of its broad-spectrum and long-lasting protection. However, spirotetramat resistance has been detected in some pests, such as *Bemisia tabaci* Gennadius [7,8], *Aphis gossypii* Glover [9], and *Phenacoccus solenopsis* [10].

High fecundity, short life cycle [11], and acaricide abuse favored evolution of resistance in *P. citri* to most acaricides in China [12–15] and other countries, such as Turkey [16]. Commercial application of spirotetramat against *P. citri* may accelerate resistance development. Unlike spirodiclofen, spirotetramat is less considered as an acaricide and its resistance management may become a serious challenge.

Selected laboratory colonies showing high resistance to a toxin may provide valuable knowledge to advance management strategies [17,18]. In the current study, a spirotetramat-resistant strain was obtained from a susceptible laboratory colony by repeated exposure to spirotetramat and then used as a tool for evaluating the risk of spirotetramat resistance in citrus red mites. We mainly examined the following aspects: (1) the dynamics of resistance to spirotetramat through laboratory selection; (2) the presence of spirotetramat resistance in different developmental stages of the mite; (3) cross-resistance between spirotetramat and other acaricides; and (4) the stability of spirotetramat resistance when individuals from the susceptible and resistant colonies were interbred.

## 2. Materials and Methods

### 2.1. Acaricides and Other Chemicals

Three ketoenol acaricides (group 23 in IRAC), spirodiclofen (Envidor®, 240 g [AI]/liter SC), spirotetramat (Movento®, 224 g [AI]/liter SC), and spiromesifen (Oberon®, 240 g [AI]/liter SC) were products of Bayer CropScience (Hangzhou, China). Other acaricides were purchased, including hexythiazox (group 10A, Nissorun®, 5% EC, Nippon Soda Co. (Tokyo, Japan)), clofentezine (group 10A, Apollo®, 200 g [AI]/liter SC, Aventis Co.)(Weifang, China), fenpyroximate (group 21A, Bamanling®, 50 g [AI]/liter SC, Nihon Nohyaku Co. (Tokyo, Japan)), etoxazole (group 10B, ZOOM®, 110 g [AI]/liter SC, Sumitomo Chemical (Tokyo, Japan)), bifenazate (group 20D, Acramite®, 430 g [AI]/liter SC, Chemtura Shanghai Co. (Shanghai, China)), abamectin (group 6, Hisun®, 1.8% EC, Zhejian Hisun Chemical Co. (Taizhou, China)), and pyridaben (group 21A, Saomanjing®, 15% EC, Kesheng Group Co. (Nanjing, China). All of the other chemicals used were obtained from Sigma-Aldrich Co., St. Louis, MO, USA.

### 2.2. Citrus Red Mite

A susceptible strain of *P. citri* (LS-FJ) has been reared on seedlings of *Citrus aurantium* without exposure to pesticides since 1998. All experiments were conducted in the laboratory under the following conditions: $25 \pm 1\,^\circ$C, $70 \pm 5\%$ RH, and 16 h light/8 h dark photoperiod. Strain LS-FJ was used as the parent generation for spirotetramat resistance selection. The detached leaf method was used for spirotetramat selection and a bioassay was conducted using the detached leaf method [15,19]. About twenty females were allowed to oviposit for 1 day on a fully expanded detached leaf, which was washed thoroughly and placed onto a 3 mm layer of water-saturated sponge in a glass Petri dish (9 cm in diameter). The detached leaves did not dry out under these conditions, which therefore allows the mites to successfully reach adulthood.

For the screening of spirotetramat resistance in *P. citri*, about 10,000 eggs produced by strain LS-FJ females were continuously exposed to the acaricide. Eggs were used for resistance selection because application of the spirotetramat at this stage mainly exhibited excellent control efficacy [6] and the eggs have an incubation period of about 8 days. Twenty females were allowed to oviposit on every detached leaf as described above. We tested the toxicity of spirotetramat every three generations, and a dose approximately equal to the $LC_{90}$ dose of the last selected generation, as determined by a Potter spray tower bioassay (described below), was used in the next three generations. The highest concentration used was $9000\ \text{mg}\cdot\text{L}^{-1}$ spirotetramat containing 0.1% Triton and the obtained spirotetramat-resistant strain was designated as ST-NK.

### 2.3. Larval and Egg Bioassays

For the egg and larval bioassays, 20 adult females were transferred to untreated leaf discs as described above and allowed to oviposit for 1 day. These eggs and larvae were used in subsequent tests for the Potter spray tower bioassay. Each test used from six to eight acaricide concentrations (including a zero dose) using a base of distilled water + 0.1% Triton X-100. A volume of 1.5 mL of the solution was applied using a Potter Precision Spray Tower (Burkard, Rickmansworth, UK) [20,21] at 1 bar for 2.30 s, resulting in a wet deposit of $1.5 \pm 0.5$ mg·cm$^{-2}$ [22]. For the ovicidal test, observations of immature mites and unhatched eggs were recorded at on day 8. Furthermore, for larval bioassay, the living mites were scored when adults were observed in the control. Mites that could not walk normally when probed with a small brush were recorded as dead.

### 2.4. Cross-Resistance Studies

Cross-resistance to ten other acaricides in ST-NK strain was tested in the ST-NK strain. Ovicidal and larvicidal activities were tested for every acaricide. The egg and larval bioassay method used for these acaricides was the same as was described in Section 2.3.

### 2.5. Crossing Experiment and Toxicological Test (Bioassay–Reciprocal Crossing LS-FJ♀× ST-NK♂and ST-NK♀× LS-FJ♂)

#### 2.5.1. Crossing Experiment

Reciprocal $F_1$ (LS-FJ♀× ST-NK♂and ST-NK♀× LS-FJ♂), backcross $F_2$ (LS-FJ♀× ST-NK♂) ♀× LS-FJ♂and (ST-NK♀× LS-FJ♂) ♀× ST-NK♂), haploid $F_1$, and haploid $F_2$ were obtained, according to the principles of arrhenotokous parthenogenesis in spider mites which determine that males develop from unfertilized eggs (haploid) and females develop from fertilized eggs (diploid) [20]. Teleiochrysalid females or males were selected for the crossing experiments. A teleiochrysalid female of the ST-NK or LS-FJ strain and about 20 adult males from LS-FJ or ST-NK were introduced to the same detached citrus leaf disks, copulated, and laid F1 eggs for 6 h. The reciprocal $F_1$ teleiochrysalid females were allowed to mate with males from LS-FJ or ST-NK. Then, a backcrossed $F_2$ progeny was obtained. The male haploid $F_1$ offspring was produced by unmated females from the LS-FJ and ST-NK strains and the male haploid $F_2$ offspring was produced by unmated females from the reciprocal crosses.

#### 2.5.2. Toxicological Test

For all crossing experiments, larval bioassays were used, as described in Section 2.3. Hatched larvae from eggs in citrus discs were directly sprayed with several concentrations of spirotetramat for the toxicological test.

The formula of Stone [23] was used to estimate the degree of dominance (D) for the $F_1$ females:

$$D = (2X_2 - X_1 - X_3)/(X_1 - X_3)$$

where $X_1$ is the log of the LC$_{50}$ of the ST-NK strain, $X_2$ is the log of the LC$_{50}$ of the (corrected) heterogeneous strains, and $X_3$ is the log of the LC$_{50}$ of the LS-FJ strain. The D values ranged from $-1$ (fully recessive resistance) to $+1$ (fully dominant resistance), and a value of 0 meant no dominance.

Expected concentration–mortality curves for the $F_2$ progeny female and haploid $F_2$ progeny larvae in the spirotetramat test were estimated under the assumption of a single major gene, as described by Georghiou [24]. Single major gene inheritance is characterized graphically by a plateau at 50% mortality of the concentration–mortality curves. Whether the observed mortalities fit the expected C–M curves was tested using the chi-square test with the following formula:

$$c = 0.5W \text{ (parent 1)} + 0.5W \text{ (parent 2)}$$

where *c* is the expected response at a given concentration and *W* is the observed response as estimated from the respective probit regression line of the parental types at a given concentration. A $\chi^2$ goodness-of-fit analysis was then used to determine how well the observed responses fitted expected responses.

### 2.6. Data Analysis

Bioassay data were analyzed according to Abbott's formula [25] using the probit analysis program POLO (LeOra Software, Berkeley, CA, USA) to estimate $LC_{50}$, $LC_{90}$, and 95% confidence limits. Resistance ratios (RRs) were calculated by dividing the $LC_{50}$ value of the ST-NK strain by the $LC_{50}$ value of the LS-FJ strain. RR = 1, no resistance; RR = 2–10, low resistance; RR = 11–40, moderate resistance; RR = 41–160, high resistance; RR > 160, very high resistance [26].

### 3. Results

#### 3.1. Susceptibilities of LS-FJ and Selected ST-NK Strain Stability

The data presented in Figure 1 show the development of resistance in the response of the LS-FJ citrus red mite strain of under selection of spirotetramat. A spirotetramat-resistant strain (ST-NK) obtained after continuous selections through 15 selection cycles (45 generations) exhibited a 1668.4-fold resistance when compared to the LS-FJ strain. The figure shows that resistance development may divided into three stages: first, slow development from 1 to 9 selection cycles; second, gradual development from 10 to 13 selection cycles; and a sudden rise appearing in the last two selection cycles. The $LC_{50}$ values rose from approximately 1 to 100 mg $L^{-1}$ spirotetramat in 12 selections.

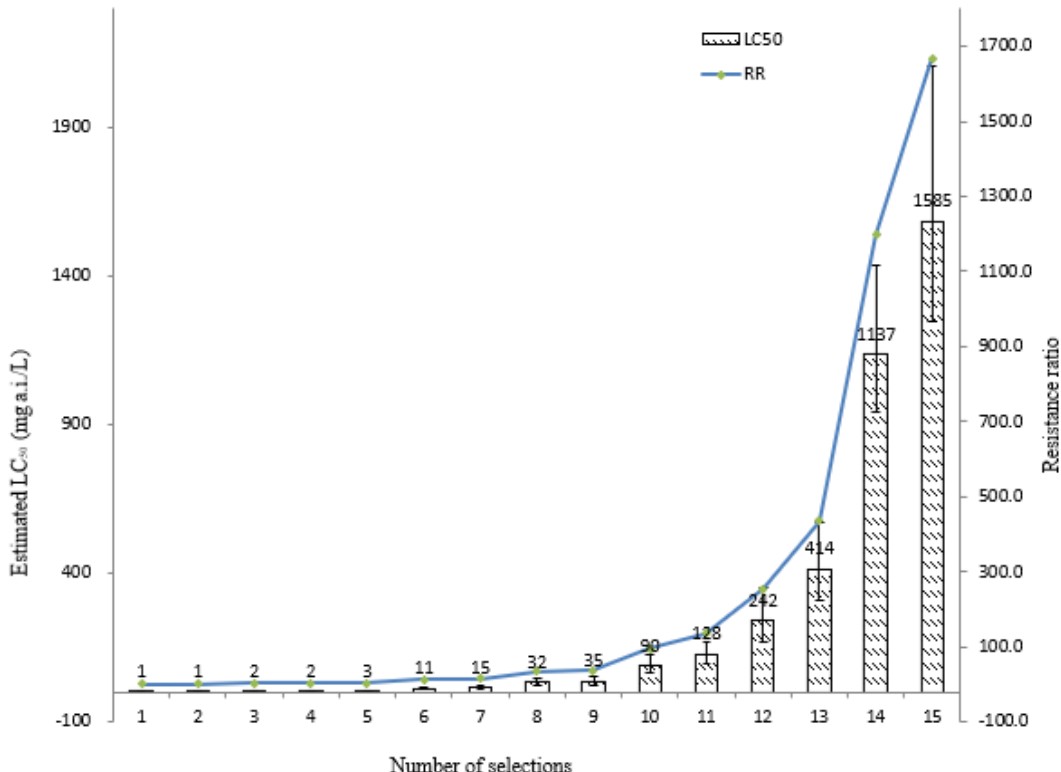

**Figure 1.** Changes in estimated $LC_{50}$ values of spirotetramat for the LS-FJ *P. citri* strain over 15 selection cycles. Detail $LC_{50}$ values are shown on each bar.

The $LC_{50}$s of the spirotetramat concentration–mortality curves were 0.95 mg $L^{-1}$ (95% CL 0.89–1.01) for the LS-FJ strain and 1589.01 mg $L^{-1}$ (95% CL 1355.94–1860.78) for the ST-NK (Figure 1) strain. An $LC_{90}$ value of 7036.69 mg $L^{-1}$ (95% CL 5359.62–10179.29) was obtained for the ST-NK strain. The slope of the concentration–mortality curve decreased

from 5.25 (SE = 0.43) for the LS-FJ strain to 1.98 (SE = 0.18) for the ST-NK strain, indicating a more heterogeneous response in the laboratory-selected resistant strain.

Resistance to spirotetramat in the mite was unstable. Without the continuous selection pressure of the acaricide, the $LC_{50}$ values of the ST-NK strain decreased gradually from 1585.01 to 329.12 mg $L^{-1}$ after 11 months (Figure 2). The $LC_{50}$ values remained stable after ten months and the overlap of $LC_{50}$s was between ten months and twelve months. The slopes showed upward tendencies with decreasing $LC_{50}$ values.

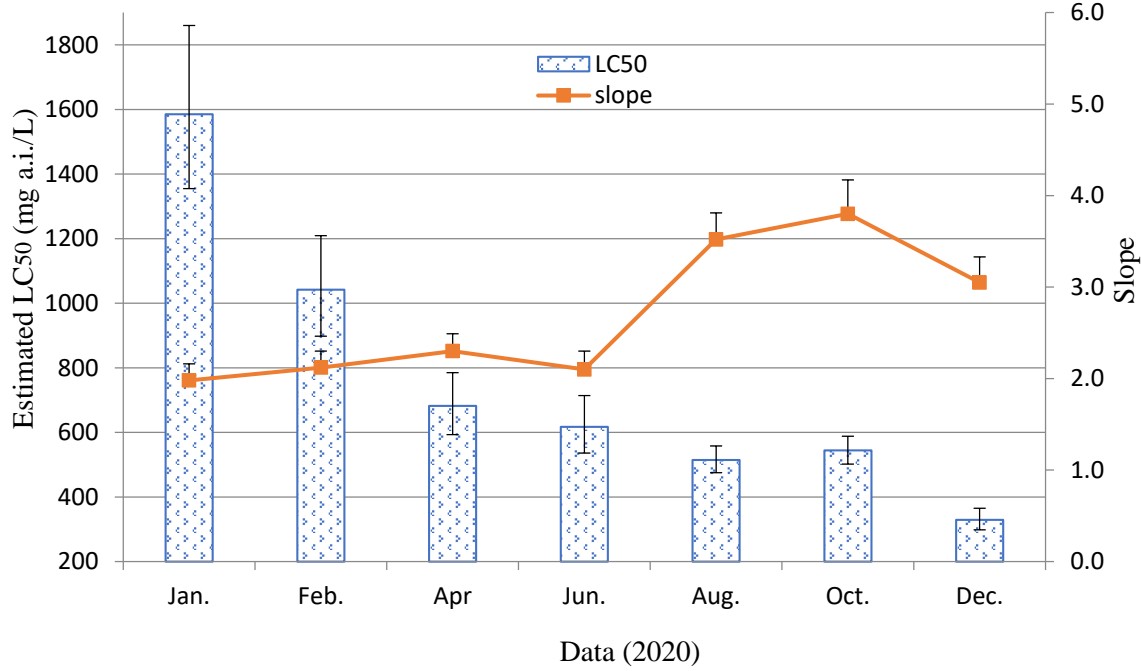

**Figure 2.** Stability of spirotetramat resistance in ST-NK mites kept on unsprayed citrus plants (starting 20 January 2020).

### 3.2. Cross-Resistance Studies

The ovicidal and larvicidal activities of eight acaricides against both the LS-FJ and ST-NK strains were tested, and cross-resistance was assessed (Table 1). Among the seven acaricides tested, the larvicidal activities of spirodiclofen and spiromesifen were relatively low against the larvae of the ST-NK strain, and the resistance ratios at $LC_{50}$ were 75.0 and 31.7, respectively. The larvae of ST-NK exhibited very low cross-resistance against hexythiazox (5.8 fold) and clofentezine (5.0 fold), but there was no cross-resistance between spirotetramat and abamectin, etoxazole, or pyridaben.

**Table 1.** Concentration probit-mortality data of different acaricides on larvae of the LS-FJ and ST-NK strains with calculated resistance ratios (RRs).

| Acaricides | | LS-FJ | | ST-NK | | |
|---|---|---|---|---|---|---|
| | | Slope (±SE) | $LC_{50}$ (95%CI) (mg $L^{-1}$) | Slope (±SE) | $LC_{50}$ (95%CI) (mg $L^{-1}$) | RR |
| Spirotetramat | Eggs | 3.41 (±0.25) | 74.15 (71.10–77.33) | 1.9 (0.22) | 10057.19 (8637.30–11,977.19) | 135.6 |
| | Larvae | 5.25 (0.43) | 0.95 (0.89–1.01) | 1.98 (0.18) | 1585.01 (1355.94–1860.78) | 1668.4 |
| Spirodiclofen | Eggs | 3.51 (0.28) | 4.57 (4.38–4.76) | 2.03 (0.18) | 163.02 (139.78–189.82) | 35.7 |
| | Larvae | 4.42 (±0.32) | 1.2 (1.12–1.29) | 2.15 (±0.19) | 90.04 (77.43–104.14) | 75.0 |
| Spiromesifen | Eggs | 2.91 (±0.23) | 2.69 (2.54–2.83) | 2.06 (±0.18) | 43.33 (37.22–50.36) | 16.1 |
| | Larvae | 4.1 (±0.31) | 0.9 (0.84–0.97) | 2.68 (±0.21) | 28.55 (24.36–33.51) | 31.7 |
| Etoxazole | Eggs | 2.56 (±0.19) | 1.38 (1.30–1.46) | 1.98 (±0.19) | 1.49 (1.26–1.76) | 1.1 |

**Table 1.** *Cont.*

| | | LS-FJ | | ST-NK | | |
| --- | --- | --- | --- | --- | --- | --- |
| **Acaricides** | | **Slope ($\pm$SE)** | **LC$_{50}$ (95%CI) (mg L$^{-1}$)** | **Slope ($\pm$SE)** | **LC$_{50}$ (95%CI) (mg L$^{-1}$)** | **RR** |
| | Larvae | 3.71 ($\pm$0.29) | 1.27 (1.17–1.38) | 2.06 ($\pm$0.16) | 1.28 (1.10–1.48) | 1.0 |
| Abamectin | Eggs | 3.94 ($\pm$0.31) | 3.04 (2.92–3.16) | 0.92 ($\pm$0.08) | 4.24 (3.67–4.91) | 1.4 |
| | Larvae | 3.72 ($\pm$0.34) | 1.31 (1.20–1.42) | 2.07 ($\pm$0.18) | 1.46 (1.25–1.70) | 1.1 |
| Pyridaben | Eggs | 2.15 ($\pm$0.19) | 4.54 (3.89–5.27) | 1.83 ($\pm$0.17) | 8.4 (7.06–9.93) | 1.9 |
| | Larvae | 2.08 ($\pm$0.18) | 5.51 (4.76–6.40) | 2.27 ($\pm$0.19) | 10.03 (8.71–11.56) | 1.8 |
| Hexythiazox | Eggs | 2.5 ($\pm$0.2) | 9.46 (8.90–10.05) | 1.04 ($\pm$0.09) | 23.01 (20.12–26.33) | 2.4 |
| | Larvae | 2.03 ($\pm$0.18) | 17.72 (15.21–20.56) | 2.1 ($\pm$0.18) | 102.33 (88.18–118.80) | 5.8 |
| Clofentezine | Eggs | 2.41 ($\pm$0.18) | 25.13 (23.66–26.61) | 1.09 ($\pm$0.09) | 50.3 (44.11–57.60) | 2.0 |
| | Larvae | 1.65 ($\pm$0.13) | 19.32 (16.28–23.02) | 2.04 ($\pm$0.18) | 95.67 (82.13–111.18) | 5.0 |

Spirotetramat exhibited the lowest ovicidal activity against LS-FJ of the eight tested acaricides and the LC$_{50}$ value was only 74.15 mg L$^{-1}$ (Table 1). Compared to the larvae, the resistance ratio of the ST-NK eggs to spirotetramat was 135.6-fold greater. For spirodiclofen and spiromesifen, the resistance ratio of the ST-NK eggs reached 35.7-fold and 16.1-fold greater than the larvae, respectively, which also showed cross-resistance between the three tetronic acid insecticides in eggs. In the case of other five tested acaricides, the ST-NK strain presented only a slightly higher LC$_{50}$ than the LS-FJ strain, indicating a lack of cross-resistance to these acaricides.

### 3.3. Mode of Inheritance of Spirotetramat Resistance

The concentration–mortality curves of the reciprocal F1 derived from the different crosses (LS-FJ♀× ST-NK♂ and ST-NK♀× LS-FJ♂), are shown in Figure 3. In the cross between the susceptible females and the spirotetramat resistance males (LS-FJ♀× ST-NK♂), the F1 line (SR + S) was drawn closer the line of the ST-NK than to that of the susceptible strain. The LC$_{50}$ values (95%CI) of the LS-FJ♀× ST-NK♂ and ST-NK♀× LS-FJ♂ reciprocal crosses were 68.81 mg L$^{-1}$ (58.18–81.05) and 91.15 mg L$^{-1}$ (78.22–106.85), respectively (Table 2). The LC$_{50}$ values of the reciprocal crosses were not significantly different, as their 95% CI overlap. This shows that spirotetramat resistance was autosomal and neither sex linkage nor maternal effects were present in the tested *P. citri* strains (Table 2). The degrees of dominance were 0.15 and 0.23 for the diploid F1 of LS-FJ♀× ST-NK♂ and ST-NK♀× LS-FJ♂, and 0.07 and 0.13 for haploid F2 of LS-FJ♀× ST-NK♂ and ST-NK♀× LS-FJ♂, respectively (Table 2). These results show that there was incompletely dominant inheritance of resistance against spirotetramat in *P. citri*.

**Table 2.** Probit statistics of the reciprocal crosses tested against spirotetramat (*n* = number of mites, D = degree of dominance).

| | **Strain** | ***n*** | **Slope ($\pm$SE)** | **LC$_{50}$ (95%CI) (mg L$^{-1}$)** | **LC$_{90}$ (95%CI) (mg L$^{-1}$)** | **D** |
| --- | --- | --- | --- | --- | --- | --- |
| Diploid F1 progeny | LS-FJ♀× ST-NK♂ | 2355 | 1.83 ($\pm$0.17) | 68.81 (58.18–81.05) | 345.79 (260.70–509.88) | 0.15 |
| | ST-NK♀× LS-FJ♂ | 2445 | 1.98 ($\pm$0.17) | 91.15 (78.22–106.85) | 416.67 (316.38–604.73) | 0.23 |
| Haploid F2 progeny | LS-FJ♀× ST-NK♂ | 3882 | 1.24 ($\pm$0.05) | 49.91 (45.00–55.06) | 544.85 (461.98–657.54) | 0.07 |
| | ST-NK♀× LS-FJ♂ | 3759 | 1.38 ($\pm$0.06) | 63.06 (56.83–69.36) | 527.91 (458.06–620.86) | 0.13 |

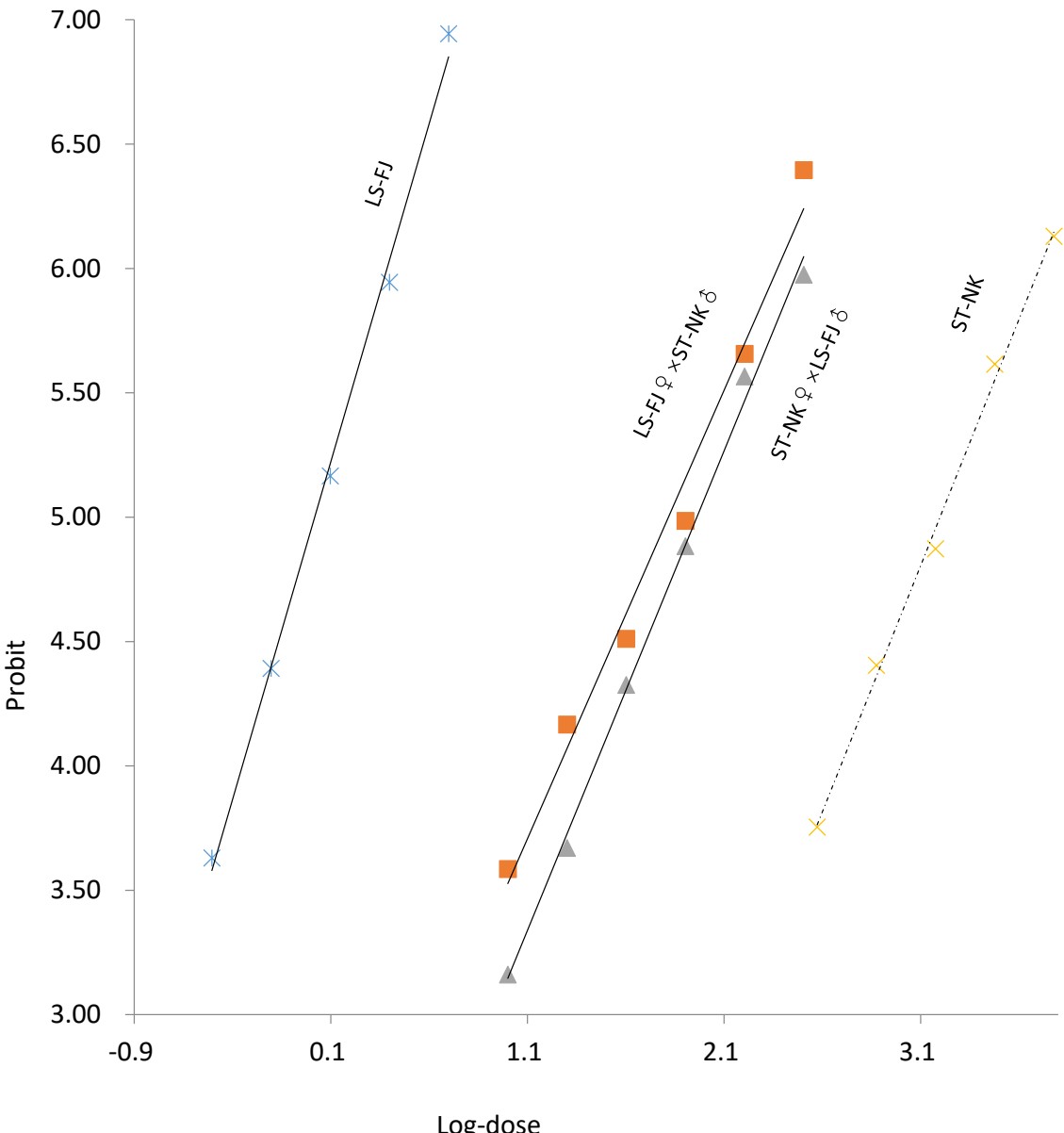

**Figure 3.** Spirotetramat concentration-mortality curves for LS-FJ, ST-NJ, and reciprocal crosses.

The log dose/probit mortality lines of spirotetramat for the backcrosses, corrected for haploid male offspring, are presented in Figure 4. The lines exhibit an absence of a plateau at 50% level or at the 75% level. On the other hand, data from the chi-squared analyses for the (LS-FJ × ST-NK) ♀× LS-FJ♂ and (ST-NK × LS-FJ) ♀× ST-NK♂(F2) were highly significant ($\chi^2$ = 98.85, df = 18 and $\chi^2$ = 249.82, df = 20, respectively, $p < 0.001$). Generally, large values of the $\chi^2$ statistic are considered to be an indication that the assumptions made about the mode of inheritance (in this case, one major gene mode) are wrong, so these results also seem to support the inheritance of multiple genes.

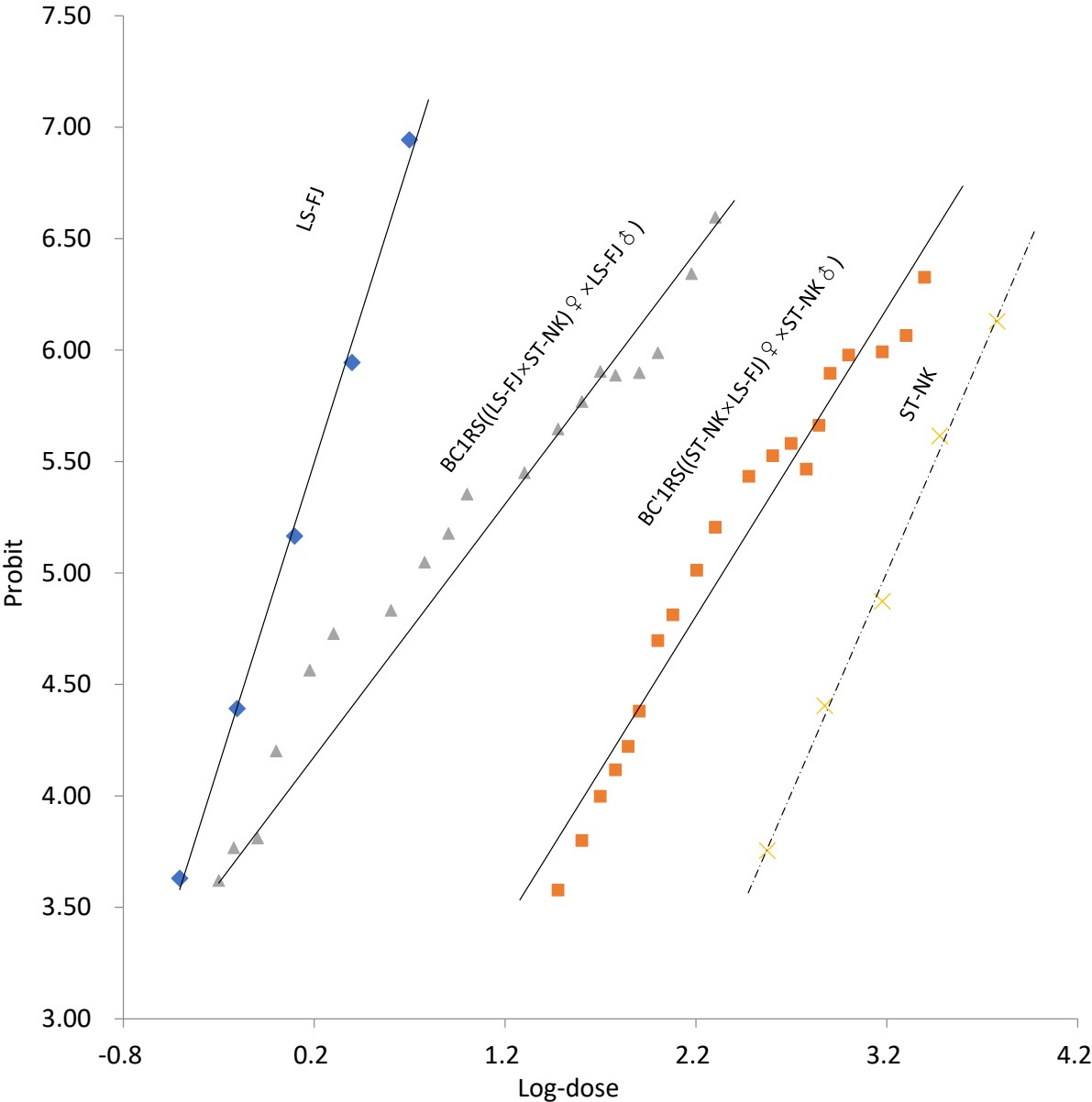

**Figure 4.** LD-P lines of actual mortality and expected mortality for backcross $BC_{1SR}$ and $BC_{1RS}$ of *Panonychus citri*.

The log dose/probit mortality lines of spirotetramat for the haploid $F_1$ males of strains LS-FJ and ST-NK and the haploid $F_2$ males of the virgin $F_1$ heterozygous females are shown in Figure 5. The line indicates the expected mortality if the single gene conferred the spirotetramat resistance. Because (1) there is an absence of a plateau at the 50% level for both crosses (Figure 5) and (2) the observed mortalities in four concentrations were significantly different ($\chi^2$ =241.62, df = 16 for LS-FJ♀× ST-NK♂, and $\chi^2$ = 139.02, df = 16 for ST-NK♀× LS-FJ♂, *p* < 0.001) compared to the expected mortalities, these results provide evidence that the inheritance of spirotetramat resistance was incompletely dominant and polygenic.

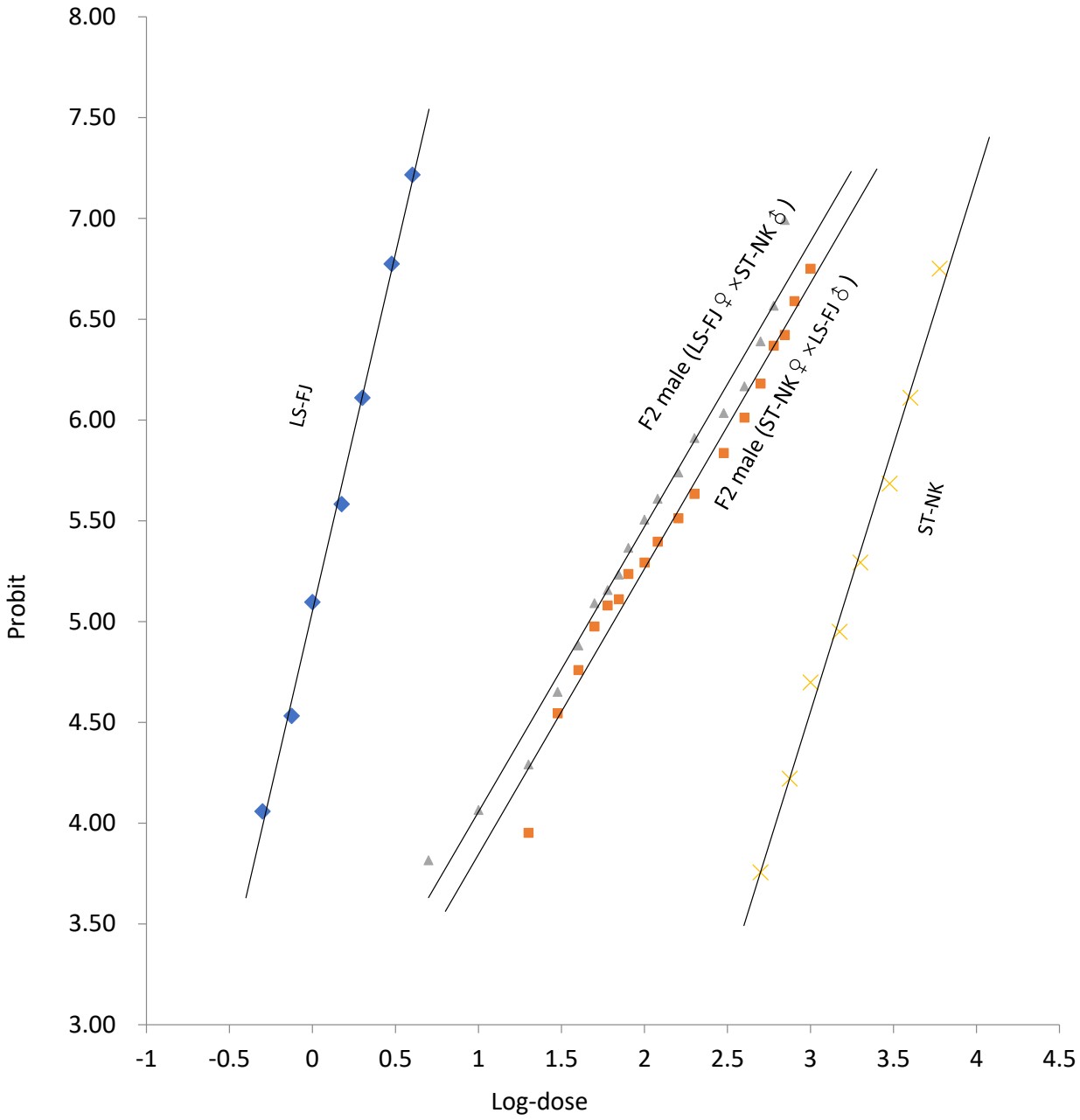

**Figure 5.** Spirotetramat concentration-mortality curves for parental males and F2 males derived from virgin F1 females.

## 4. Discussion

After continuous exposure to spirotetramat for 45 generations, the ST-NK population of *P. citri* developed a 1668.4-fold and 135.6-fold resistance to spirotetramat in larvae and eggs, respectively. The resistance development of spirotetramat in the mite seemed low when compared with other acaricides, such as hexythiazox and spirodiclofen. Yamamoto et al. [19], after selection of the Haibara strain of citrus red mite with hexythiazox for three generations, found that this strain exhibited 23,000-fold resistance. We screened the same susceptible strain (LS-FJ) with spirodiclofen for 42 generations and got > 1000-fold resistance) [27]. Furthermore, highly spirotetramat-resistant populations were also obtained in sucking pests, such as *Aphis gossypii* (50 generations for 441.26-fold resistance) [9] and *Phenacoccus solenopsis* (13 generations for 328.69-fold resistance) [10] by artificial selection in the laboratory. All these examples illustrate that some pests, including *P. citri*, have the potential to develop a high level of resistance to spirotetramat; although



the resistance level obtained by artificial selection in the laboratory was usually equaled or exceeded in field populations within one decade [28].

In the absence of spirotetramat, resistance has reverted from an original high of 1668.4-fold to 346.4-fold. This is similar to the results from a study by Ejaz Masood and Ali Shad Safraz et al. [10], who also noted spirotetramat resistance to be generally unstable in spirotetramat resistant *P. solenopsis*. Furthermore, unstable resistance was also reported in resistance to spirodiclofen [20] or spiromesifen [29] in *Tetranychus urticae* Koch. The instability of spirocyclic tetramic acid derivative resistance in mites and other pests may be favorable for resistance management. Based on this study, without spirotetramat application for one year, the $LC_{50}$ value was 329.1 mg $L^{-1}$, much higher than the field-recommended concentration (56 mg $L^{-1}$) against *P. citri*, the discriminating concentration (31.6 mg $L^{-1}$) [6], and the concentration recommended by Bayer CropScience (44.8–56 mg $L^{-1}$) [3]. For resistance management tactics, the rotation time of spirotetramat with other different mode of action acaricides should be more than one year at this resistance level. However, populations in the field may be affected by many factors and genetic exchange is inevitable. For example, the reversion of hexythiazox resistance in *P. citri* might be influenced by the quantity of susceptible individuals immigrating successively [30].

Spider mite development differs somewhat between species, and the life cycle is composed of the egg, the larva, two nymphal stages (protonymph and deutonymph), and the adult. Unlike spirodiclofen and spiromesifen, which exhibited excellent action on eggs and immature mites, spirotetramat had a relatively low efficacy in preventing the hatch of mite eggs and low acute toxicity against immature mites. Ouyang et al. [6] recommended an 8-day egg bioassay and a 11-day egg bioassay for spirodiclofen and spirotetramat resistance monitoring in *P. citri*, respectively, but occurrence of resistance of the two acaricides in eggs may be not assured. Previous studies documented that spirodiclofen resistance usually occurred in the larvae, but not the eggs of *T. urticae* [20] and *Panonychus ulmi* [31,32]. The present research indicates that highly spirotetramat resistance exists in both the eggs and larvae of *P. citri*. Furthermore, cross-resistance to spirodiclofen and spiromesifen occurred in both the eggs and larvae of *P. citri*. This difference may be caused by the choice of life stage in which *P. citri* is exposed to the acaricide. Eggs were used for selecting in this study, while larvae were used in *T. urticae* because a low resistance was also observed in larvae-selected spirodiclofen-resistant *P. citri* [27]. In the field, mite generations overlap, with different life stages present at the same time; thus, egg resistance must be considered when resistance management strategies are first being developed and applied.

Spirotetramat cross-resistance to other acaricides has not been reported in highly spirotetramat-resistant mites. The present results show that the ST-NK of *P. citri* (140-fold) developed very low cross-resistance to hexythiazox and clofentezine (95% CI did not overlap) while no cross-resistance to etoxazole, abamectin and pyridaben (95% CI overlap). We also reported that multi-resistant field populations exhibited high resistances to spirodiclofen, abamectin, clofentezine, pyridaben, and hexythiazox and a low resistance to spirotetramat in the absence of spirotetramat [15]. According to their effect mechanisms, spirodiclofen, spirotetramat, etoxazole, spirodiclofen, and hexythiazox are classified as growth inhibitors and abamectin is classified as a neurotoxin acaricides [33]. Regarding other pests, multiresistant *M. persicae* (organophosphates, carbamates, and pyrethroids) and *B. tabaci* (organophosphates, neonicotinoids, pyrethroids, buprofezin, pyriproxyfen, and pymetrozin) strains showed that no cross-resistance occurred between spirotetramat and other conventional insecticides [5,34]. However, a laboratory-selected spirotetramat-resistant strain (SR) of cotton aphid also conferred high-level cross-resistance to alpha-cypermethrin and bifenthrin [8]. The Spiro-SEL population *Phenacoccus solenopsis* had a moderate level of cross-resistance with profenofos (26.01-fold RR) and bifenthrin (28.04-fold RR) and a high level of cross-resistance with abamectin (45.37-fold RR) when compared to the lab-PK population [10].

Reciprocal crossing between resistant and susceptible strains provides information on resistant genes, such as whether they are dominant or recessive, sex-linked, or autoso-

mal [35]. The results of bioassays after reciprocal crosses between LS-FJ and ST-NK showed that there was no significant difference in the $LC_{50}$ between the F1 progeny, indicating that resistance to spirotetramat is inherited autosomally. Unlike resistance to most kinds of developmental inhibitors acaricides, such as hexythiazox (incompletely recessive, eggs) [19], etoxazole (completely recessive in *T. urticae*) [36], and clofentezine (recessive) [37], which is incompletely recessive, the resistance to spirotetramat in *P. citri* was incompletely dominant and autosomal. In this aspect, it was similar to spirodiclofen resistance in *T. urticae* [20]. Croft and Van de Bann [38] concluded that recessive resistance tends to be less stable in the field and provides the greatest opportunity to manage resistance.

Assuming that spirotetramat resistance in *P. citri* is controlled by two or more genes, the resistance would be likely to spread more slowly than if the resistance was mediated by a single gene [39]. Like most mites, *P. citri* also have arrhenotokous parthenogenesis [40] which may favor the fixation of fewer but more advantageous alleles [41] and also lead to contradiction of genetic mode of many acaricide's resistance under different selection conditions, such as regional practices. For example, chlorfenapyr resistance in Japanese populations of *T. urticae* [36] is under monogenic control, while in Belgian populations, it is polygenic [42]. Monogenic resistance is favored under field selection regimes [39] and acaricide resistance is typically monogenic [19,43]. Resistance levels under monogenic control are usually much lower than those under polygenic control. The $LC_{50}$ for hexythiazox is notably higher for polygenic resistance in Japan ($10,000\,\mathrm{mg\,L^{-1}}$) [44] than for monogenic resistance in Australia ($48\,\mathrm{mg\,L^{-1}}$) [43], whereas similar $LC_{50}$ values for chlorfenapyr were reported in a Japanese resistant population ($2130\,\mathrm{mg\,L^{-1}}$) [36] and a Belgian resistant population ($2939\,\mathrm{mg\,L^{-1}}$) [42]. If this conflict occurred in the spirotetramat-related genes of *P. citri*, genetic analysis of field spirotetramat-resistant population would be necessary, which may provide much more comprehensive information to manage spirotetramat resistance.

## 5. Conclusions

In conclusion, a high potential for the development of resistance to spirotetramat was observed in *P. citri*, and high resistance levels were observed not only in eggs, but also in larvae. The instability of resistance without spirotetramat means that the spirotetramat may possibly be reused after its withdrawal for some time. Spirotetramat exhibited cross-resistance with spirodiclofen and spiromesifen, but not with some other conventional acaricides. Spirotetramat resistance in *P. citri* was found to be under polygenic control and incompletely dominant inheritance, which create difficulty in the management of resistance.

**Author Contributions:** J.H. and G.F. conceived, designed, and performed the experiments and data analyses. J.W., Y.Y. and C.W. collected the data and performed the analysis. F.C. and W.R. helped perform the analysis with constructive discussions. G.F. is the lead author. All authors contributed to manuscript preparation. All authors have read and agreed to the published version of the manuscript.

**Funding:** This work was supported by the Chinese National Natural Science Foundation (31201547), the Fujian Provincial Department of Science & Technology (2015R1024-9, 2016R1023-7, 2020R10240011), and the Science and Technology Innovation Foundation of FAAS Supported by Financial Department of Fujian Government (CXTD2021002-1, XTCXGC2021017).

**Institutional Review Board Statement:** Not applicable.

**Informed Consent Statement:** Not applicable.

**Data Availability Statement:** Data are available from the corresponding author.

**Acknowledgments:** We would like to thank Lin Lei and Fei Shuhua for their support with rearing mites.

**Conflicts of Interest:** The authors declare no conflict of interest.

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
