# Peer review of "Cross-Resistance Pattern and Genetic Studies in Spirotetramat-Resistant Citrus Red Mite, Panonychus citri (Acari: Tetranychidae)"

_agriculture, doi:10.3390/agriculture12050737_

Round 1

Reviewer 1 Report

This is a review of the manuscript entitled “Cross-resistance pattern and genetic studies in spirotetramat-resistant citrus red mite, Panonychus citri (Acari: Tetranychidae). The paper is somewhat improved over the first submission but problems in the presentation persist. The mistakes make the paper harder to read as they are distracting but generally not substantive.

Line 14) “…, but not to 5 other acaricides.” At least change others to other.

Line 16) …was autosomal with neither sex …”

Line 19) …incompletely dominant.

The problems with English persist. The problems are annoying but generally do not affect the communication of scientific content.

Line 25) producer (singular) not producers (plural). China is a country (singular) not China are countries (plural).

Line 25) … with 2.8 million planted hectares producing 41.4 million tons [1].

Line 30) kill the not killthe

Line 30) About 1319 acaricides’ products …   The possessive does not work. Acaricide is modifying products. A better choice is: About 1319 acaricide products,

Line 31) maybe “registered for the”  However, I am not clear on the goal in this sentence. Are there 1319 acaricide products registered for all agricultural production in China, or just citrus? Of that number only 4% contain spirotetramat. Being such a small fraction of the available products, does this even matter?

Line 39) Movento ® (spirotetramat) was first registered in 2010 for scale insect management, but citrus red mite was added to the label in 2016.

Line 43) has not have

Line 44)  and Phenacoccus solenopsis [10].

Line 45) ebolution? No such word in English language.

Line 85) True, but not useful. The application of any pesticide accelerates resistance to that pesticide. A pesticide that is never used might lose efficacy through cross-resistance but otherwise will maintain efficacy for as long as it is never used.

Line 88) delete “much more” as the emphasis has no real content.

Line 90) … obtained from a susceptible laboratory colony by repeated exposure to spirotetramat and then used as a tool for evaluating the risk of spirotetramat resistance in the citrus red mite.

Line 95) 4) the stability of spirotetramat resistance when individuals from the susceptible and resistant colonies were interbred.

Line 99) Three ketoenol acaricides

102) purchased

Line 117) ion?

Line 117) if there was no exposure to pesticides (line 112), why wash the leaf?

Line 121-122) The first sentence is about screening and the next sentence is about selection. What parts of the following sentences are about screening and which ones are about selection?

  1. I need to test the current generation with several doses to estimate the LC90
  2. I need to select a different set of mites using the estimated LC90.

Line 122) continuous applications? Did you really set up an automatic spray system that kept spraying the mites with pesticide every second of every day? That would be continuous application. An alternative is that they were exposed to freshly treated leaves.. In part, I am asking if it was continuous application or continuous exposure.

Line 123) The justifications don’t work for me.

  1. The egg stage was used for resistance selection because mortality was high? Why is this better than using a slightly higher dose and treating larvae?
  2. The egg stage was used because it lasts about 8 days. Why does this matter when the survivors must complete their entire lifecycle to give you the next generation?
  3. It is good to know that the egg stage was used to select for resistance. I need to know this because another paper might have different results using a different life stage.

Line 126) “and approximately …” could be improved by rewording to something like “at the estimated LC90 dose for the previous generation using the bioassay described below.”

Line 132) above not abov

Line 133) “the Potter spray tower”

Line 133) respectively to what? “These eggs and the resulting larvae were used in subsequent tests.”

Line 133) “Each test was replicated …”       Each test used six to eight acaricide concentrations (including a zero dose) using a base of distilled water + 0.1% Triton X-100.

(My line numbers jump from 136 to 416)

417) Observations should not be capitalized.

Line 419) Mites that could not walk normally when prodded with a small brush were recorded as dead.

Line 423) Cross-resistance to ten other acaricides was tested in the ST-NK strain.

Line 442) “was used to estimate” does not work in the same sentence as “was estimated”. Please reword.

Line 461) Why would you use Abbott’s formula after corrected mortalities were less than 5%? As written, this is wrong. What was really done?

Line 515-519) This just describes an exponential curve.

Line 526) In scientific writing, do not start a paragraph with “however.”

Line 529) In Figure 2, it looks like January is month 1. Therefore, November is month 11. There is no month 12. With no overlap in error bars between Aug and Nov, it looks like LC50 values are continuing to decline. There is no indication of stability as suggested in Line 128. The figure legend states the start was January, so January is time 0, and the figure ends 10 months later. There is no data after 10 months and stating “stability” is entirely fictitious.  

Figure 2) All months abbreviate with three letters: Aug, Nov.

Figure 2) The figure is slightly deceptive in that the interval between bars is not constant. This is a problem to be avoided whenever possible.

Figure 2) The error bars for Apr, and Jun are unclear due to the overlap in the two graphs. Can you shift the graph of slope over ten pixels? That should separate the error bars while keeping the points within the LC50 bars.

Line 553) Check the journal policy on numbers in the text. Often small numbers like seven are written out.

Line 564) FJstrain?

Line 572) Fig. 3

Fig 3) labels should be 1 line and next to the line. LS-FJ is two lines in my version.

Line 635) The paragraph seems disjointed consisting of several ideas smashed together with insufficient connections to enable a reader to reconstruct the whole. Each sentence is useful, they just don’t quite go together in this way. The last two sentences are worthless as written.

  1. Spirotetramat resistance in this experiment (Line 630)
  2. Compare this with other acaricides (Line 632)
    1. Hexythiazox
    2. Spirodiclofen
  3. Compare to other resistance
    1. Aphis gossypii
    2. Phenacoccus solenopsis
  4. Artificial selection in the laboratory appears in the field.
  5. This method selects for polygenic resistance.
  6. This method is not entirely representative of natural resistance development.
  7. Artificial selection can give us information.
  8. All pests have the potential to develop resistance.

Line 637) generations

Line 661) recommended

Line 662) tactics

Line 662) mode of action

Line 664) transition to the hexythiazox example needs work.

Line 674) assure of what? Or assurance of what?

Line 679) Citing the unpublished work does not help this paper.

Line 682) … must be considered when developing a resistance management strategy.

Line 698) aphidalso

Line 735) controlledby

Line 737) mites,

Line 748) I have no idea what this sentence means. If the conflict occurs why is field selection necessary?

Author Response

We would like to thank the referee again for taking the time to review our manuscript.

Point 1: Line 14) “…, but not to 5 other acaricides.” At least change others to other.

Response 1: We changed “others” to “other”.

Point 2: Line 16) …was autosomal with neither sex …”

Response 2: We changed the sentence.

Point 3: Line 19) …incompletely dominant.

Response 3: We changed “dominance” to “dominant”.

The problems with English persist. The problems are annoying but generally do not affect the communication of scientific content.

Point 4: Line 25) producer (singular) not producers (plural). China is a country (singular) not China are countries (plural).

Response 4: We changed “producers” to “producer”.

Point 5: Line 25) … with 2.8 million planted hectares producing 41.4 million tons [1].

Response 5: We changed the sentence as “with 2.8 million planted hectares producing 41.4 million tons”.

Point 6: Line 30) kill the not killthe

Response 6: We changed “killthe” to “kill the”.

Point 7: Line 30) About 1319 acaricides’ products …   The possessive does not work. Acaricide is modifying products. A better choice is: About 1319 acaricide products,

Response 7: We changed “1319 acaricides’ products” to “1319 acaricide products”.

Point 8: Line 31) maybe “registered for the”  However, I am not clear on the goal in this sentence. Are there 1319 acaricide products registered for all agricultural production in China, or just citrus? Of that number only 4% contain spirotetramat. Being such a small fraction of the available products, does this even matter?

Response 8: We want to emphasis the number of acaricide products and there are relative less spirotetramat products because of spirotetramat patent protection. In the future, there will be more products containing spirotetramat.

Point 9: Line 39) Movento ® (spirotetramat) was first registered in 2010 for scale insect management, but citrus red mite was added to the label in 2016.

Response 9: We changed the sentence to “Movento ® (spirotetramat) was first registered in 2010 for scale insect management, but citrus red mite was added to the label in 2016.”.

Point 10: Line 43) has not have

Response 10: “have” was changed to “has”

Point 11: Line 44)  and Phenacoccus solenopsis [10].

Response 11: Add “and” before “Phenacoccus solenopsis ”.

Point 12: Line 45) ebolution? No such word in English language.

Response 12: We changed “ebolution” to “evolution”.

Point 13: Line 85) True, but not useful. The application of any pesticide accelerates resistance to that pesticide. A pesticide that is never used might lose efficacy through cross-resistance but otherwise will maintain efficacy for as long as it is never used.

Response 13: It may provide some information useful for delaying the resistance development.

Point 14: Line 88) delete “much more” as the emphasis has no real content.

Response 14: In my revision, “much more” in line 51 was deleted.

Point 15: Line 90) … obtained from a susceptible laboratory colony by repeated exposure to spirotetramat and then used as a tool for evaluating the risk of spirotetramat resistance in the citrus red mite.

Response 15: We changed the sentence according to the suggestion.

Point 16: Line 95) 4) the stability of spirotetramat resistance when individuals from the susceptible and resistant colonies were interbred.

Response 16: We changed the sentence.

Point 17: Line 99) Three ketoenol acaricides

Response 17: We changed “ketoenols” to “ketoenol”.

Point 18: 102) purchased

Response 18: We changed “purchase” to “purchased”.

Point 19: Line 117) ion?

Response 19: We changed “ion” to “on”.

Point 20: Line 117) if there was no exposure to pesticides (line 112), why wash the leaf?

Response 20: There may be some dust.

Point 21: Line 121-122) The first sentence is about screening and the next sentence is about selection. What parts of the following sentences are about screening and which ones are about selection?

  1. I need to test the current generation with several doses to estimate the LC90
  2. I need to select a different set of mites using the estimated LC90.

Response 21: we tested current generation with several doses to estimate the LC90 which used as selected dose used in the nest three generations. And we added “in the nest three generations”

Point 22: Line 122) continuous applications? Did you really set up an automatic spray system that kept spraying the mites with pesticide every second of every day? That would be continuous application. An alternative is that they were exposed to freshly treated leaves.. In part, I am asking if it was continuous application or continuous exposure.

Response 22: We changed “applications” to “exposure”.

Point 23: Line 123) The justifications don’t work for me.

  1. The egg stage was used for resistance selection because mortality was high? Why is this better than using a slightly higher dose and treating larvae?
  2. The egg stage was used because it lasts about 8 days. Why does this matter when the survivors must complete their entire lifecycle to give you the next generation?
  3. It is good to know that the egg stage was used to select for resistance. I need to know this because another paper might have different results using a different life stage.

Response 23: The method was also recommended for spirotetramat bioassay on the mite by Bayer Crop science.

Point 24: Line 126) “and approximately …” could be improved by rewording to something like “at the Point : estimated LC90 dose for the previous generation using the bioassay described below.”

Response 24: We changed the sentence.

Point 25: Line 132) above not abov

Response 25: We changed to “above”.

Point 26: Line 133) “the Potter spray tower”

Response 26: We change to “the Potter spray tower”.

Point 27: Line 133) respectively to what? “These eggs and the resulting larvae were used in subsequent tests.”

Response 27: We added “in subsequent tests” to the sentence.

Point 28: Line 133) “Each test was replicated …”       Each test used six to eight acaricide concentrations (including a zero dose) using a base of distilled water + 0.1% Triton X-100.

Response 28: We revised the sentence to “Each test used six to eight acaricide concentrations (including a zero dose) using a base of distilled water + 0.1% Triton X-100.”

(My line numbers jump from 136 to 416)

Point 29: 417) Observations should not be capitalized.

Response 29: We changed to “observation”.

Point 30: Line 419) Mites that could not walk normally when prodded with a small brush were recorded as dead.

Response 30: We revised the sentence as the suggestion.

Point 31: Line 423) Cross-resistance to ten other acaricides was tested in the ST-NK strain.

Response 31: We revised the sentence as the suggestion.

Point 32: Line 442) “was used to estimate” does not work in the same sentence as “was estimated”. Please reword.

Response 32: We changed “estimate” to “calculate”.

Point 33: Line 461) Why would you use Abbott’s formula after corrected mortalities were less than 5%? As written, this is wrong. What was really done?

Response 33: We deleted “after corrected mortalities were less than 5%”.

Point 34: Line 515-519) This just describes an exponential curve.

Response 34: We just meant that.

Point 35: Line 526) In scientific writing, do not start a paragraph with “however.”

Response 35: We deleted the word “however”.

Point 36: Line 529) In Figure 2, it looks like January is month 1. Therefore, November is month 11. There is no month 12. With no overlap in error bars between Aug and Nov, it looks like LC50 values are continuing to decline. There is no indication of stability as suggested in Line 128. The figure legend states the start was January, so January is time 0, and the figure ends 10 months later. There is no data after 10 months and stating “stability” is entirely fictitious.  

Response 36: The experiment of stability ended in December 2020. We revised “November” to “December” and 12 months to 11 months.

Point 37: Figure 2) All months abbreviate with three letters: Aug, Nov.

Response 37: We revised that.

Point 38: Figure 2) The figure is slightly deceptive in that the interval between bars is not constant. This is a problem to be avoided whenever possible.

Response 38: We revised that.

Point 39: Figure 2) The error bars for Apr, and Jun are unclear due to the overlap in the two graphs. Can you shift the graph of slope over ten pixels? That should separate the error bars while keeping the points within the LC50 bars.

Response 39: We revised that.

Point 40: Line 553) Check the journal policy on numbers in the text. Often small numbers like seven are written out.

Response 40: We changed “7” to “seven”.

Point 41: Line 564) FJstrain?

Response 41: We changed “LS-FJstrain” to “LS-FJ strain”.

Point 42: Line 572) Fig. 3

Response 42: We changed “Figs. 3” to " Fig. 3”.

Point 43: Fig 3) labels should be 1 line and next to the line. LS-FJ is two lines in my version.

Response 43: We revised that.

Point 44: Line 635) The paragraph seems disjointed consisting of several ideas smashed together with insufficient connections to enable a reader to reconstruct the whole. Each sentence is useful, they just don’t quite go together in this way. The last two sentences are worthless as written.

  1. Spirotetramat resistance in this experiment (Line 630)
  2. Compare this with other acaricides (Line 632)
    1. Hexythiazox
    2. Spirodiclofen
  3. Compare to other resistance
    1. Aphis gossypii
    2. Phenacoccus solenopsis
  4. Artificial selection in the laboratory appears in the field.
  5. This method selects for polygenic resistance.
  6. This method is not entirely representative of natural resistance development.
  7. Artificial selection can give us information.
  8. All pests have the potential to develop resistance.

Response 44: We deleted two sentence.

Point 45: Line 637) generations

Response 45: We changed “gnerations” to “generations”.

Point 46: Line 661) recommended

Response 46: We changed “recommende” to “recommended”.

Point 47: Line 662) tactics

Response 47: We changed “tactic” to tactics “”.

Point 48: Line 662) mode of action

Response 48: We added “mode of”.

Point 49: Line 664) transition to the hexythiazox example needs work.

Response 49: We added “For example,”.

Point 50: Line 674) assure of what? Or assurance of what?

Response 50: “occurrence of resistance of spirotetramat and spirodiclofen in eggs”

Point 51: Line 679) Citing the unpublished work does not help this paper.

Response 51: We deleted the sentence.

Point 52: Line 682) … must be considered when developing a resistance management strategy.

Response 52: We revised the sentence.

Point 53: Line 698) aphidalso

Response 53: We changed to “aphid also”.

Point 54: Line 735) controlledby

Response 54: We changed to “controlled by”.

Point 55: Line 737) mites,

Response 55: We changed “mite” to mites.

Point 56: Line 748) I have no idea what this sentence means. If the conflict occurs why is field selection necessary?

Response 56: We revised the sentence to “If this confict occurred in spirotetramat genetic of P. citri, genetic analysis of field spirotetramat-reistant population would be necessary, which may provide much more comprehevsive information to manage spirotetramat resistance.”.

Reviewer 2 Report

There are still some minor edits to be done. 

Author Response

We would like to thank the referee again for taking the time to review our manuscript.

There are still some minor edits to be done. 

Reviewer 3 Report

This primary study is impactful in monitoring and management of citrus red mite resistance to insecticides. However, the scientific writing of the manuscript needs to improve a lot. The author should rewrite the manuscript more scientifically. Careful attention should provide to the language of the manuscript. The manuscript is difficult to read. Please, Check the grammatical and lexical errors strictly and carefully. 

Author Response

This primary study is impactful in monitoring and management of citrus red mite resistance to insecticides. However, the scientific writing of the manuscript needs to improve a lot. The author should rewrite the manuscript more scientifically. Careful attention should provide to the language of the manuscript. The manuscript is difficult to read. Please, Check the grammatical and lexical errors strictly and carefully. 

Response: We have revised a little of grammatical and lexical errors.

Round 2

Reviewer 3 Report

This manuscript is interesting and has practical application n monitoring and management of citrus red mite resistance. The authors have attempted to make some corrections in the revised version but still, there are some important issues that need to be revised. The article still has serious grammatical mistakes. Extensive language editing is required. My suggestion is to revise the manuscript by a scientific English writing expert.

Other comments are provided below:

Line 10: Correct spelling “Spirotetramt-resistant”.

Line 17-18: What did you mean by “diploid F1” and “haploid F2”? Describe in the method section how you developed “diploid F1” and “haploid F2”?, and how did you confirm the ploidy level/

Line 22: Correct spelling “Rresistance”.

Line 30-32: Rewrite the sentence clearly.

Line 71: Correct spelling “indibiduals”.

Line 86: the word “reared” should be added between “been” and “on”.

Line 90: Replace “were cultured” with “was conducted”, and “use of” with “using”.

Line 100-103: Split the sentence and write clearly.

Line 190: Correct spelling “Spirotetramt-resistant”.

Line 243-244: How have you determined that spirotetramat resistance was autosomal, not sex-linked maternally inherited? Make it clear.

Line 244: There is no Table 3 in the manuscript. Correct the table number.

Line 254: Give a space between “level” and “or”.

Line 324-326: Rewrite the sentence clearly.

Line 335: “The different may be caused by the selected life stage”. Make clear what you want to say.

Line 337-339: Split the sentence and write clearly.

Line 382: Correct spelling “contradiction”.

Line 396: Make italic “P. citri”.

Author Response

We would like to thank the referee again for taking the time to review our manuscript.

This manuscript is interesting and has practical application n monitoring and management of citrus red mite resistance. The authors have attempted to make some corrections in the revised version but still, there are some important issues that need to be revised. The article still has serious grammatical mistakes. Extensive language editing is required. My suggestion is to revise the manuscript by a scientific English writing expert.

We revised some grammatical mistakes.

Other comments are provided below:

Point 1: Line 10: Correct spelling “Spirotetramt-resistant”.

Response 1: We changed “Spirotetramt-resistant” to “Spirotetramat-resistant”.

Point 2: Line 17-18: What did you mean by “diploid F1” and “haploid F2”? Describe in the method section how you developed “diploid F1” and “haploid F2”?, and how did you confirm the ploidy level/

Response 2: We added “according to arrhenotokous parthenogenesis in spider mite that males develop from unfertilized eggs (haploid), and females develop from fertilized eggs (diploid) [20].” at line 120.

Point 3: Line 22: Correct spelling “Rresistance”.

Response 3: We changed “Rresistance” to “Resistance”.

Point 4: Line 30-32: Rewrite the sentence clearly.

Response 4: We rewrote the sentence as “About 1319 acaricide products have been registered in the pest management in the country, among which 53 products contained spirotetramat[3]”.

Point 5: Line 71: Correct spelling “indibiduals”.

Response 5: We changed “individuals” to “indibiduals”.

Point 6: Line 86: the word “reared” should be added between “been” and “on”.

Responsed 6: We added “reared”.

Point 7: Line 90: Replace “were cultured” with “was conducted”, and “use of” with “using”.

Response 7: We replaced “were cultured” with “was conducted”, and “use of” with “using”.

Point 8: Line 100-103: Split the sentence and write clearly.

Response 8: We changed to “Spirotetramat used in experiment was prepared with deionized water containing 0.1% Triton-100. All Three replicates of six to seven concentrations of spirotetramat plus a control (deionized water) were tested. Leaves treated with distilled water containing Triton X-100 alone were used as the control.”

Point 9: Line 190: Correct spelling “Spirotetramt-resistant”.

Response 9: We changed to “Spirotetramat-resistant”.

Point 10: Line 243-244: How have you determined that spirotetramat resistance was autosomal, not sex-linked maternally inherited? Make it clear.

Response 10: We judged that based on “LC50 of LS-FJ♀×ST-NK♂ and ST-NK♀×LS-FJ ♂reciprocal crosses were not significantly different as their 95 % CI overlap”.

Point 11: Line 244: There is no Table 3 in the manuscript. Correct the table number.

Response 11: We change to “Table 2”.

Point 12: Line 254: Give a space between “level” and “or”. 222

Response 12: We added a space.

Point 13: Line 324-326: Rewrite the sentence clearly. 292

Response 13: We changed to “The present research indicated high spirotetramat resistance existed in both eggs and larvae of P. citri. Furthermore, cross-resistance to spirodiclofen and spiromesifen also occurred both in eggs and larvae of P. citri.:

Point 14: Line 335: “The different may be caused by the selected life stage”. Make clear what you want to say.

Response 14: We changed to “The difference may be caused by the chosen life stage exposed the acaricide.”

Point 15: Line 337-339: Split the sentence and write clearly.

Response 15: We changed to “In the field, mite generations overlap, with different life stages present at the same time; thus, egg resistance must be considered when resistance management strategies are first being developed and applied.”

Point 16: Line 382: Correct spelling “contradiction”.

Respone 16: We changed “contadiction” to “contradiction”.

Point 17: Line 396: Make italic “P. citri”.

Response 17:. We change that.

This manuscript is a resubmission of an earlier submission. The following is a list of the peer review reports and author responses from that submission.

Round 1

Reviewer 1 Report

This is a review for the manuscript entitled “Genetic analysis and cross-resistance spectrum of a laboratory-selected spirotetramat resistant strain of citrus red mite (Acari: Tetranychidae).” This is an important topic as the problem plagues all of agriculture. Studies such as this one provide more insight into pesticide resistance and how we might manage it for this pest and others.

In the first paragraph, I can tell that English is a problem. Issues include subject-verb agreement and using plural versus singular forms of words. The wording can also be more direct. The first paragraph should read something like this:

China is the world’s largest citrus producer, with 2.8 million planted hectares and 41.4 million tons of production [1]. The citrus red mite, Panonychus citri (McGregor) (Acari: Tetranychidae), is one of the most important citrus pests, mainly occurring in the spring and autumn. Citrus red mite feeding habits result in defoliation, twig dieback, fruit drop, and generally poor tree health and reduced production. Chinese citrus growers depend mostly on the application of acaricides to suppress the pest [2]. About 1319 acaricide products have been registered for mite suppression in China, of which 53 contain spirotetramat [3].

Line 36) What are manoes?

Line 38) activity not active

Line 38) mites not mite

The authors need to clean up the writing, but I will ignore most instances of this problem except where it causes problems with understanding.

Line 45) The cited article does not appear to support the claim for arrhenotokous reproduction.

Line 45) Arrhenotokous reproduction can take several forms. Which one is relevant here?

Line 47) turkey is a bird native to North America in the genus Meleagris. In contrast, Turkey is a country in Western Asia – Southeastern Europe.

Line 47) Resistance to …   this sentence does not make sense.

Line 58) You selected mites for spirotetramat resistance but now have a spirodiclofen resistant strain. How does that work?

Line 63) Please organize these by mode of action. One example is IRAC classification: https://irac.org .  

Line 93) What is a “detached citrus Potter Precision Spray Tower”? I know about Potter towers, just not sure how the detached citrus modification to a Potter tower works. I think what you mean is that detached citrus leaves were placed into a Potter Precision Spray Tower, but that is just my guess.

Line 95) A mature citrus leaf can be several years old. How did you ensure that these leaves were never exposed to pesticides? Even greenhouse-grown plants sometimes need pest management, though that might only be an insecticidal soap or spray oil.

Line 97) glass or plastic Petri dishes?

Line 102) Why distilled water? In some places, growers will add adjuvants to the spray tank. Wetter/spreader surfactants are commonly used. Do growers in China ever use such things or is distilled water more typical of farm practice?

Line 103) Would not the application rate depend on flow rate, and flow rate on formulation? You are not applying pure water, so for a fixed pressure and duration, the application rate will vary.

Line 104) The cited work was entitled “The acaricidal effect of sulfur ….” I fail to see how that would support your contention that a 2.30 second application at 1 bar delivers 1.5 (+- 0.5) mg/cm.

Line 105) Adding surfactant only to the control was a mistake.

Line 110) How did you define “walk normally?” This is not the criterion I am used to in assessing mortality.

Line 152) It has been a long time since I used POLO. Were all the graphs and analyses carried out using this program?

  Figure 3) Use negative values rather than parentheses (or explain the meaning of values in red).

Figure 3) align text closer to the relevant regression line.

Line 229) I don’t see a dotted line in the figure.

Figure 5) Are you sure the regression lines are labeled correctly? My initial thought was that ST-NK and LS-FJ should be reversed as it is in all the other graphs.

Line 251) Is this the “same susceptible strain” as used in this study or as used by Yamamoto et al?

Line 256-259) repeated material.

Line 261) information about what exactly? Just finish the thought, I am not disagreeing.

Line 278) That would have been interesting. Given your resistant population, what happens if the resistant individuals are exposed to some number of LS-FJ individuals? Maybe a future experiment.

Line 345) absevered?

Line 346) meat?

Author Response

Thank very much for carefully and professional review.

Point1: In the first paragraph, I can tell that English is a problem. Issues include subject-verb agreement and using plural versus singular forms of words. The wording can also be more direct. The first paragraph should read something like this:

China is the world’s largest citrus producer, with 2.8 million planted hectares and 41.4 million tons of production [1]. The citrus red mite, Panonychus citri (McGregor) (Acari: Tetranychidae), is one of the most important citrus pests, mainly occurring in the spring and autumn. Citrus red mite feeding habits result in defoliation, twig dieback, fruit drop, and generally poor tree health and reduced production. Chinese citrus growers depend mostly on the application of acaricides to suppress the pest [2]. About 1319 acaricide products have been registered for mite suppression in China, of which 53 contain spirotetramat [3].

Response1: We revised the first paragraph.

Point 2: Line 36) What are manoes?

Response 2: “manoes” changed to “mangoes”.

Point 3: Line 38) activity not active

Response 3: “active” changed to “activity”.

Point4: Line 38) mites not mite

Response 4: “mite” changed to “mites”.

Point 5: The authors need to clean up the writing, but I will ignore most instances of this problem except where it causes problems with understanding.

Response 5: We revised.

Point 6: Line 45) The cited article does not appear to support the claim for arrhenotokous reproduction.

Response 6: “arrhenotokous reproduction” changed to “short life cycle”

Point 7: Line 45) Arrhenotokous reproduction can take several forms. Which one is relevant here?

Response 7: “arrhenotokous reproduction” changed to “short life cycle”

Point 8: Line 47) turkey is a bird native to North America in the genus Meleagris. In contrast, Turkey is a country in Western Asia – Southeastern Europe.

Response 8: “turkey” changed to “Turkey”

Point 8: Line 47) Resistance to …   this sentence does not make sense.

Response 8: We deleted the sentence.

Point 9: Line 58) You selected mites for spirotetramat resistance but now have a spirodiclofen resistant strain. How does that work?

Response 9: We also selected a spirodiclofen resistant strain of the mite in the laboratory.

Point 10: Line 63) Please organize these by mode of action. One example is IRAC classification: https://irac.org .  

Response 10: We have added the classification.

Point 11: Line 93) What is a “detached citrus Potter Precision Spray Tower”? I know about Potter towers, just not sure how the detached citrus modification to a Potter tower works. I think what you mean is that detached citrus leaves were placed into a Potter Precision Spray Tower, but that is just my guess.

Response 11: We wrote the method again.

Point 12: Line 95) A mature citrus leaf can be several years old. How did you ensure that these leaves were never exposed to pesticides? Even greenhouse-grown plants sometimes need pest management, though that might only be an insecticidal soap or spray oil.

Response 12: The leaves were obtained from the seedlings.

Point 13: Line 97) glass or plastic Petri dishes?

Response 13: glass Petri dishes

Point 14: Line 102) Why distilled water? In some places, growers will add adjuvants to the spray tank. Wetter/spreader surfactants are commonly used. Do growers in China ever use such things or is distilled water more typical of farm practice?

Response 14: We wanted to reduce the effect of water in laboratory.

Point 15: Line 103) Would not the application rate depend on flow rate, and flow rate on formulation? You are not applying pure water, so for a fixed pressure and duration, the application rate will vary.

Response 15: It need to fix the volume of acaricide and pressure in Spray Tower bioassay.

Point 16: Line 104) The cited work was entitled “The acaricidal effect of sulfur ….” I fail to see how that would support your contention that a 2.30 second application at 1 bar delivers 1.5 (+- 0.5) mg/cm.

Response 16: We wrote these again.

Point 17: Line 105) Adding surfactant only to the control was a mistake.

Response 17: We added “acaricides (prepared with distilled water + 0.1% Triton X-100)”

Point 18: Line 110) How did you define “walk normally?” This is not the criterion I am used to in assessing mortality.

Response 18: It is a criterion usually used in mite bioassay.

Point 19: Line 152) It has been a long time since I used POLO. Were all the graphs and analyses carried out using this program?

Response 19: Yes, we did.

Point 20:  Figure 3) Use negative values rather than parentheses (or explain the meaning of values in red).

Response 20: We have changed to negative values.

Point 21: Figure 3) align text closer to the relevant regression line.

Response 21: We did that.

Point 22: Line 229) I don’t see a dotted line in the figure.

Response 22: We detected “dotted”

Point 23: Figure 5) Are you sure the regression lines are labeled correctly? My initial thought was that ST-NK and LS-FJ should be reversed as it is in all the other graphs.

Response23: We exchanged labels of ST-NK and LS-FJ.

Point 24: Line 251) Is this the “same susceptible strain” as used in this study or as used by Yamamoto et al?

Response 24: We added “LS-FJ”.

Point 25: Line 256-259) repeated material.

Response 25: We wrote these against.

Point 26: Line 261) information about what exactly? Just finish the thought, I am not disagreeing.

Response 26: We change into “All these examples illustrate that some pests, including P. citri, have potential to develop high level of resistance to spirotetramat.”

Point 27: Line 278) That would have been interesting. Given your resistant population, what happens if the resistant individuals are exposed to some number of LS-FJ individuals? Maybe a future experiment.

Response 27: The bioassay against cross or backcross between LS-FJ and ST-NK may provide the evidence.

Point 28: Line 345) absevered?

Response 28: “absevered” changed to “observed”.

Point 29: Line 346) meat?

Response 29: “meat” changed to “meant”.

Reviewer 2 Report

This manuscript describes a long-term selection experiment with spirotetramat using citrus red mite as the target pest. The assays that were conducted are standard practice in the field and were both adequately replicated and analyzed. Stability of resistance was measured as well as cross-resistance with spirodicofen and spiromesifen. Overall the replication conducted was sufficient, in my opinion. The assays that were conducted are standard practice in the field and were both adequately replicated and analyzed. The data are adequately interpreted and placed in the context of the literature. The conclusions are justified by the results.

Unfortunately, the biggest drawback to the presentation of the manuscript is poor English grammar in many areas as well as many unnecessary typographical errors unrelated to grammar. It's strange, because the grammar is completely fine at some points and then there are sporatic problems. I do not have the time to comprehensively edit this manuscript for writing and I think it requires significant re-writing to make it easier to read and understand. This manuscript really should be sent to a professional English grammar service for comprehensive editing. Afterwards, the authors could simply proofread their manuscript to remove the typographical errors (lack of spaces after full stop, ect.).

Section 2.5: It is difficult to understand all of these crosses based on the explanation provided here. This needs to be clarified. Perhaps a figure would clarify. Or a list of the final crosses and their purpose for the experiment should be explained. I was only able to begin to understand the reasons for these crosses, as I began to read the results. It would be useful to describe the underlying reasons for the crosses in the methods.

Author Response

Thank very much for carefully and professional review.

Point 1:nfortunately, the biggest drawback to the presentation of the manuscript is poor English grammar in many areas as well as many unnecessary typographical errors unrelated to grammar. It's strange, because the grammar is completely fine at some points and then there are sporatic problems. I do not have the time to comprehensively edit this manuscript for writing and I think it requires significant re-writing to make it easier to read and understand. This manuscript really should be sent to a professional English grammar service for comprehensive editing. Afterwards, the authors could simply proofread their manuscript to remove the typographical errors (lack of spaces after full stop, ect.).

Response 1: Sorry for our poor English writing and we wrote again to do better.

Point 2: Section 2.5: It is difficult to understand all of these crosses based on the explanation provided here. This needs to be clarified. Perhaps a figure would clarify. Or a list of the final crosses and their purpose for the experiment should be explained. I was only able to begin to understand the reasons for these crosses, as I began to read the results. It would be useful to describe the underlying reasons for the crosses in the methods.

Response 2: We have written the method again. In genetic analyses, both the cross and backcross are necessary.

Reviewer 3 Report

The authors investigated spirotetramat resistance in the citrus red mite by selecting for resistance from a pesticide naïve laboratory strain by sequentially increasing selection pressure. The data from the study is well analyzed and presented, although some parts of the manuscript is very verbose e.g., the abstract.

An issue to be addressed in this study is that the toxicity of abamectin, and pyridaben was tested on egg and larva mites. These miticides are often evaluated for their adulticide efficacy. The authors need to address this in the method and discussion sessions.

Selection with low dose typically select for minor resistance genes that often result in unstable resistance phenotypes when the selection pressure is over. This is often not the case on the field when high doses are used to typically achieve ~ 90% efficacy and consequently selecting for resistance via a single major locus. The authors need to discuss how the selection regime in the lab might not align with citrus mite acaricide resistance patten in field scenarios.

There are plethora of grammatical errors and typos.

Line 58: replace spirodiclofen with spirotetramat

Line 114: change lara to larval

Line 304: Spirodiclofen is a lipid biosynthesis inhibitor.

Line 282: Change effective to efficacy

Author Response

Thank very much for carefully and professional review.

Point 1: The authors investigated spirotetramat resistance in the citrus red mite by selecting for resistance from a pesticide naïve laboratory strain by sequentially increasing selection pressure. The data from the study is well analyzed and presented, although some parts of the manuscript is very verbose e.g., the abstract.

Response 1: We wrote the abstract and materials again.

Point 2: An issue to be addressed in this study is that the toxicity of abamectin, and pyridaben was tested on egg and larva mites. These miticides are often evaluated for their adulticide efficacy. The authors need to address this in the method and discussion sessions.

Response 2: We tested ovicidal and larvicidal activity mainly because of the spirotetramat mainly act on developmental stage of the mite and abamectin and pyridaben also exhibited activies.

Point 3: Selection with low dose typically select for minor resistance genes that often result in unstable resistance phenotypes when the selection pressure is over. This is often not the case on the field when high doses are used to typically achieve ~ 90% efficacy and consequently selecting for resistance via a single major locus. The authors need to discuss how the selection regime in the lab might not align with citrus mite acaricide resistance patten in field scenarios.

 Response 3: We discussed the effect of environment on acaricide resistance genetic only based on other studies, but not based on our results

Point 4:There are plethora of grammatical errors and typos.

Response 4: We tried to correct the grammatical errors.

Point 5: Line 58: replace spirodiclofen with spirotetramat

Response 5: We replaced “spirodiclofen” with “spirotetramat”.

Point 6: Line 114: change lara to larval

Response 6: We replaced “lara” with “larval”.

Point 7: Line 304: Spirodiclofen is a lipid biosynthesis inhibitor.

Response 7: We meant spirodiclofen may inhibit the development of the mite.

Point 8: Line 282: Change effective to efficacy

Response 8: We replaced “effective” with “efficacy”.